# Fair Algorithms for Clustering

**Suman K. Bera**
UC Santa Cruz
Santa Cruz, CA 95064
sbera@ucsc.edu

**Deeparnab Chakrabarty**
Dartmouth College
Hanover, NH 03755
deeparnab@dartmouth.edu

**Nicolas J. Flores**
Dartmouth College
Hanover, NH 03755
nicolasflores.19@dartmouth.edu

**Maryam Negahbani**
Dartmouth College
Hanover, NH 03755
maryam@cs.dartmouth.edu

## Abstract

We study the problem of finding low-cost *fair clusterings* in data where each data point may belong to many protected groups. Our work significantly generalizes the seminal work of Chierichetti et al. (NIPS 2017) as follows.

- We allow the user to specify the parameters that define fair representation. More precisely, these parameters define the maximum over- and minimum under-representation of any group in any cluster.
- Our clustering algorithm works on any $\ell_p$-norm objective (e.g. $k$-means, $k$-median, and $k$-center). Indeed, our algorithm transforms any vanilla clustering solution into a fair one incurring only a slight loss in quality.
- Our algorithm also allows individuals to lie in multiple protected groups. In other words, we do not need the protected groups to partition the data and we can maintain fairness across different groups simultaneously.

Our experiments show that on established data sets, our algorithm performs much better in practice than what our theoretical results suggest.

## 1   Introduction

Many important decisions today are made by machine learning algorithms. These range from showing advertisements to customers [49, 23], to awarding home loans [38, 46], to predicting recidivism [6, 24, 21]. It is important to ensure that such algorithms are *fair* and are not biased towards or against some specific groups in the population. A considerable amount of work [37, 57, 19, 36, 15, 56, 55] addressing this issue has emerged in the recent years.

Our paper considers fair algorithms for clustering. Clustering is a fundamental unsupervised learning problem where one wants to partition a given data-set. In machine learning, clustering is often used for feature generation and enhancement as well. It is thus important to consider the bias and unfairness issues when inspecting the quality of clusters. The question of fairness in clustering was first asked in the beautiful paper of Chierichetti et al. [19] with subsequent generalizations by Rösner and Schmidt [50].

*In this paper, we give a much more generalized and tunable notion of fairness in clustering than that in [19, 50]. Our main result is that any solution for a wide suite of vanilla clustering objectives can be transformed into fair solutions in our notion with only a slight loss in quality by a simple algorithm.*

Many works in fairness [15, 19, 50, 14] work within the disparate impact (DI) doctrine [28]. Broadly speaking, the doctrine posits that any "protected class" must have approximately equal representation in the decisions taken (by an algorithm). Although the DI doctrine is a law [1, 27] in the United States, violating the DI doctrine is by itself *not* illegal [3]; it is illegal only if the violation cannot be justified by the decision maker. In the clustering setting, this translates to the following algorithmic question : what is the loss in quality of the clustering when all protected classes are required to have approximately equal representation in the clusters returned?

Motivated thus, Chierichetti et al. [19], and later Rösner and Schmidt [50], model the set of points as partitioned into $\ell$ colors, and the color proportion of each returned cluster should be similar to that in the original data. There are three shortcomings of these papers: (a) the fairness constraint was too stringent and brittle, (b) good algorithms were given only for the $k$-center objective, and (c) the color classes weren't allowed to overlap. We remark that the last restriction is limiting since an individual can lie in multiple protected classes (consider an African-American senior woman). In our work we address all these concerns: we allow the user to specify the fairness constraints, we give simple algorithms with provable theoretical guarantees for a large suite of objective functions, and we allow overlapping protected classes.

**Our fairness notion.** We propose a model which extends the model of [19] to have $\ell \geq 2$ groups of people which are allowed to overlap. For each group $i$, we have two parameters $\beta_i, \alpha_i \in [0, 1]$. Motivated by the DI doctrine, we deem a clustering solution *fair* if each cluster satisfies two properties: (a) *restricted dominance (RD)*, which asserts that the fraction of people from group $i$ in any cluster is at most $\alpha_i$, and (b) *minority protection (MP)*, which asserts that the fraction of people from group $i$ in any cluster is at least $\beta_i$. Note that we allow $\beta_i, \alpha_i$'s to be arbitrary parameters, and furthermore, they can differ across different groups. This allows our model to provide a lot of flexibility to users. For instance, our model easily captures the notions defined by [19] and [50].

We allow our protected groups to overlap. Nevertheless, the quality of our solutions depend on the amount of overlap. We define $\Delta$ (similar to [15]) to be the maximum number of groups a single individual can be a part of. This parameter, as we argued above, is usually not 1, but can be assumed to be a small constant depending on the application.

**Our results.** Despite the generality of our model, we show that *in a black-box fashion*, we can get fair algorithms for *any $\ell_p$-norm objective* (this includes, $k$-center, $k$-median, and the widely used $k$-means objective) if we allow for very small additive violations to the fairness constraint. We show that *given any $\rho$-approximation algorithm $\mathcal{A}$ for a given objective which could be returning widely unfair clusters, we can return a solution which is a $(\rho + 2)$-approximation to the best clustering which satisfies the fairness constraints* (Theorem 1). Our solution, however, can violate both the RD and MP property *additively* by $4\Delta + 3$. This is negligible if the clusters are large, and our empirical results show this almost never exceeds 3. Further in our experiments, our cost is at most 15% more than optimum, which is a much better factor compared to $(\rho + 2)$.

The black-box feature of our result is useful also in *comparing* the performance of any particular algorithm $\mathcal{A}$. This helps if one wishes to justify the property of an algorithm one might be already using. Our results can be interpreted to give a way to convert any clustering algorithm to its fair version. Indeed, our method is very simple – we use the solution returned by $\mathcal{A}$ to define a *fair assignment* problem and show that this problem has a good optimal solution. The fair assignment problem is then solved via iterative rounding which leads to the small additive violations. In the case of $\Delta = 1$ (disjoint groups), we can get a simpler, one-iteration rounding algorithm.

**Comparison with recent works.** In a very recent independent and concurrent work, Schmidt et al. [51] consider the fair $k$-means problem in the streaming model with a notion of fairness similar to ours. However, their results crucially assume that the underlying metric space is Euclidean. Their main contributions are defining "fair coresets" and showing how to compute them in a streaming setting, resulting in significant reduction in the input size. Although their coreset construction algorithm works with arbitrary number of groups, their fair $k$-means algorithms assume there are only two disjoint groups of equal size. Even for this, Schmidt et al. [51] give an $(5.5\rho + 1)$-approximation, given any $\rho$-approximation for the vanilla $k$-means problem; the reader should compare with our $(\rho + 2)$-approximation. Backurs et al. [9] consider the problem of designing scalable algorithm for the fair k-median problem in the Euclidean space. The notion of fairness is *balance*, as defined by Chierichetti et al. [19], and hence works only for two disjoint groups. Their approximation ratio is $O_{r,b}(d \log n)$ where $r$ and $b$ are fairness parameters, and $d$ is the dimension of the Euclidean space. In

contrast, our fair $k$-means and $k$-median algorithms works in any metric space, with arbitrary number of overlapping groups. Independently and concurrently, Ahmadian et al. [4] study the fair k-center problem with only the *restricted dominance* constraints, and Bercea et al. [10] consider a variety of clustering objectives in a fairness model that is similar to ours. Both of these works give similar, but arguably more complicated algorithms with similar theoretical guarantees as ours. In comparison, we emphasize on a simple, yet powerful unifying framework that can handle any $\ell_p$-norm objective. None of these above works handle overlapping groups.

## 1.1 Other related works

Fairness in algorithm design has received a lot of attention lately [12, 45, 26, 28, 37, 57, 19, 36, 15, 56, 55, 15, 14, 22, 40, 29]. Our work falls in the category of designing fair algorithms, and as mentioned, we concentrate on the notion of *disparate impact*. Feldman et al. [28] and Zafar et al. [56] study the fair classification problem under this notion. Celis et al. in [15], Celis et al. in [14], and Chierichetti et al. in [20] study respectively the fair ranking problem, the multiwinner voting problem, and the matroid optimization problem; All of these works model fairness through *disparate impact*. Chierichetti et al. in [19] first addresses *disparate impact* for clustering problems in the presence of two groups, Rösner and Schmidt [50] generalizes it to more than two groups.

Chen et al. [18] define a notion of proportionally fair clustering where all possible groups of reasonably large size are entitled to choose a center for themselves. This work builds on the assumption that sometimes the task of identifying protected group itself is untenable. Kleindessner et al. in [41] study the problem of enforcing fair representation in the data points chosen as cluster center. This problem can also be posed as a matroid center problem. Kleindessner et al. in [42] extends the fairness notion to graph spectral clustering problems. Celis et al. in [13] proposes a meta algorithm for the classification problem under a large class of fairness constraints with respect to multiple non-disjoint protected groups.

Clustering is a ubiquitous problem and has been extensively studied in diverse communities (see [2] for a recent survey). We focus on the work done in the algorithms and optimization community for clustering problems under $\ell_p$ norms. The $p = \{1, 2, \infty\}$ norms, that is the $k$-median, $k$-means, and $k$-center problems respectively, have been extensively studied. The $k$-center problem has a 2-approximation [31, 30] and it is NP-hard to do better [32]. A suite of algorithms [17, 35, 16, 8, 44] for the $k$-median problem has culminated in a 2.676-approximation [11], and is still an active area of research. For $k$-means, the best algorithm is a $9 + \varepsilon$-approximation due to Ahmadian et al. [5]. For the general $p$-norm, most of the $k$-median algorithms imply a constant approximation.

## 2 Preliminaries

Let $C$ be a set of points (whom we also call "clients") we want to cluster. Let these points be embedded in a metric space $(\mathcal{X}, d)$. We let $F \subseteq \mathcal{X}$ be the set of possible cluster center locations (whom we also call "facilities"). Note $F$ and $C$ needn't be disjoint, and indeed $F$ could be equal to $C$. For a set $S \subseteq \mathcal{X}$ and a point $x \in \mathcal{X}$, we use $d(x, S)$ to denote $\min_{y \in S} d(x, y)$. For an integer $n$, we use $[n]$ to denote the set $\{1, 2, \ldots, n\}$.

Given the metric space $(\mathcal{X}, d)$ and an integer parameter $k$, in the VANILLA $(k, p)$-CLUSTERING problem the objective is to (a) *"open"* a subset $S \subseteq F$ of at most $k$ facilities, and (b) find an *assignment* $\phi : C \to S$ of clients to open facilities so as to minimize $\mathcal{L}_p(S; \phi) := \left( \sum_{v \in C} d(v, \phi(v))^p \right)^{\frac{1}{p}}$. Indeed, in this vanilla version with no fairness considerations, every point $v \in C$ would be assigned to the closest center in $S$. The case of $p = \{1, 2, \infty\}$, the $k$-median, $k$-means, and $k$-center problems respectively, have been extensively studied in the literature [31, 30, 17, 35, 16, 8, 44, 11, 5]. Given an instance $\mathcal{I}$ of the VANILLA $(k, p)$-CLUSTERING problem, we use $\text{OPT}_{\text{vnll}}(\mathcal{I})$ to denote its optimal value.

The next definition formalizes the fair clustering problem which is the main focus of this paper.

**Definition 1** (FAIR $(k, p)$-CLUSTERING Problem). In the fair version of the clustering problem, one is additionally given $\ell$ many (not necessarily disjoint) *groups* of $C$, namely $C_1, C_2, \ldots, C_\ell$. We use $\Delta$ to denote the maximum number of groups a single client $v \in C$ can belong to; so if the $C_j$'s were disjoint we would have $\Delta = 1$. One is also given two *fairness vectors* $\vec{\alpha}, \vec{\beta} \in [0, 1]^\ell$.

The objective is to (a) *open* a subset of facilities $S \subseteq F$ of at most $k$ facilities, and (b) find an *assignment* $\phi : C \to S$ of clients to the open facilities so as to minimize $\mathcal{L}_p(S; \phi)$, where $\phi$ satisfies the following *fairness constraints*.

$$\left|\{v \in C_i : \phi(v) = f\}\right| \leq \alpha_i \cdot \left|\{v \in C : \phi(v) = f\}\right|, \quad \forall f \in S, \forall i \in [\ell], \qquad \text{(RD)}$$

$$\left|\{v \in C_i : \phi(v) = f\}\right| \geq \beta_i \cdot \left|\{v \in C : \phi(v) = f\}\right|, \quad \forall f \in S, \forall i \in [\ell], \qquad \text{(MP)}$$

The assignment $\phi$ defines a cluster $\{v : \phi(v) = f\}$ around every open facility $f \in S$. As explained in the Introduction, eq. (RD) is the *restricted dominance* property which upper bounds the ratio of any group's participation in a cluster, and eq. (MP) is the *minority protection* property which lower bounds this ratio to protect against under-representation. Due to these fairness constraints, we can no longer assume $\phi(v)$ is the nearest open facility in $S$ to $v$. Indeed, we use the tuple $(S, \phi)$ to denote a fair-clustering solution.

We use $\text{OPT}_{\text{fair}}(\mathcal{I})$ to denote the optimal value of any instance $\mathcal{I}$ of the FAIR $(k, p)$-CLUSTERING problem. Since $\mathcal{I}$ is also an instance of the vanilla problem, and since every fair solution is also a vanilla solution (but not necessarily vice versa) we get $\text{OPT}_{\text{vnll}}(\mathcal{I}) \leq \text{OPT}_{\text{fair}}(\mathcal{I})$ for any $\mathcal{I}$.

A fair clustering solution $(S, \phi)$ has $\lambda$-*additive* violation, if the eq. (RD) and eq. (MP) constraints are satisfied upto $\pm \lambda$-violation. More precisely, for any $f \in S$ and for any group $i \in [\ell]$, we have

$$\beta_i \cdot \left|\{v \in C : \phi(v) = f\}\right| - \lambda \leq \left|\{v \in C_i : \phi(v) = f\}\right| \leq \alpha_i \cdot \left|\{v \in C : \phi(v) = f\}\right| + \lambda \quad \text{(V)}$$

Our main result is the following.

**Theorem 1.** *Given a $\rho$-approximate algorithm $\mathcal{A}$ for the* VANILLA $(k, p)$-CLUSTERING *problem, we can return a $(\rho + 2)$-approximate solution $(S, \phi)$ with $(4\Delta + 3)$-additive violation for the* FAIR $(k, p)$-CLUSTERING *problem.*

In particular, we get $O(1)$-factor approximations to the FAIR $(k, p)$-CLUSTERING problem with $O(\Delta)$ *additive* violation, for any $\ell_p$ norm. Furthermore, for the important special case of $\Delta = 1$, our additive violation is at most $+3$.

## 3   Algorithm for the FAIR $(k, p)$-CLUSTERING problem

Our algorithm is a simple two step procedure. First, we solve the VANILLA $(k, p)$-CLUSTERING problem using some algorithm $\mathcal{A}$, and fix the centers $S$ opened by $\mathcal{A}$. Then, we solve a *fair reassignment problem*, called FAIR $p$-ASSIGNMENT problem, on the same set of facilities to get assignment $\phi$. We return $(S, \phi)$ as our fair solution.

**Definition 2** (FAIR $p$-ASSIGNMENT Problem)**.** In this problem, we are given the original set of clients $C$ and a set $S \subseteq F$ with $|S| = k$. The objective is to find the assignment $\phi : C \to S$ such that (a) the constraints eq. (RD) and eq. (MP) are satisfied, and (b) $\mathcal{L}_p(S; \phi)$ is minimized among all such satisfying assignments.

Given an instance $\mathcal{J}$ of the FAIR $p$-ASSIGNMENT problem, we let $\text{OPT}_{\text{asgn}}(\mathcal{J})$ denote its optimum value. Clearly, given any instance $\mathcal{I}$ of the FAIR $(k, p)$-CLUSTERING problem, if $S^*$ is the optimal subset for $\mathcal{I}$ and $\mathcal{J}$ is the instance of FAIR $p$-ASSIGNMENT defined by $S^*$, then $\text{OPT}_{\text{fair}}(\mathcal{I}) = \text{OPT}_{\text{asgn}}(\mathcal{J})$. A $\lambda$-violating algorithm for the FAIR $p$-ASSIGNMENT problem is allowed to incur $\lambda$-additive violation to the fairness constraints.

### 3.1   Reducing FAIR $(k, p)$-CLUSTERING to FAIR $p$-ASSIGNMENT

In this section we present a simple reduction from the FAIR $(k, p)$-CLUSTERING problem to the FAIR $p$-ASSIGNMENT problem that uses a VANILLA $(k, p)$-CLUSTERING solver as a black-box.

**Theorem 2.** *Given a $\rho$-approximate algorithm $\mathcal{A}$ for the* VANILLA $(k, p)$-CLUSTERING *problem and a $\lambda$-violating algorithm $\mathcal{B}$ for the* FAIR $p$-ASSIGNMENT *problem, there is a $(\rho + 2)$- approximation algorithm for the* FAIR $(k, p)$-CLUSTERING *problem with $\lambda$-additive violation.*

*Proof.* Given instance $\mathcal{I}$ of the FAIR $(k, p)$-CLUSTERING problem, we run $\mathcal{A}$ on $\mathcal{I}$ to get a (not-necessarily fair) solution $(S, \phi)$. We are guaranteed $\mathcal{L}_p(S; \phi) \leq \rho \cdot \text{OPT}_{\text{vnll}}(\mathcal{I}) \leq \rho \cdot \text{OPT}_{\text{fair}}(\mathcal{I})$. Let $\mathcal{J}$ be the instance of FAIR $p$-ASSIGNMENT obtained by taking $S$ as the set of facilities. We run algorithm $\mathcal{B}$ on $\mathcal{J}$ to get a $\lambda$-violating solution $\hat{\phi}$. We return $(S, \hat{\phi})$.

By definition of $\lambda$-violating solutions, we get that $(S, \hat{\phi})$ satisfies eq. (V) and that $\mathcal{L}_p(S, \hat{\phi}) \leq \text{OPT}_{\text{asgn}}(\mathcal{J})$. The proof of the theorem follows from the lemma below. □

**Lemma 3.** $\text{OPT}_{\text{asgn}}(\mathcal{J}) \leq (\rho + 2) \cdot \text{OPT}_{\text{fair}}(\mathcal{I})$.

*Proof.* Suppose the optimal solution of $\mathcal{I}$ is $(S^*, \phi^*)$ with $\mathcal{L}_p(S^*; \phi^*) = \text{OPT}_{\text{fair}}(\mathcal{I})$. Recall $(S, \phi)$ is the solution returned by the $\rho$-approximate algorithm $\mathcal{A}$. We describe the existence of an assignment $\phi' : C \to S$ such that $\phi'$ satisfies eq. (RD) and eq. (MP), and $\mathcal{L}_p(S; \phi') \leq (\rho + 2) \cdot \text{OPT}_{\text{fair}}(\mathcal{I})$. Since $\phi'$ is a feasible solution of $\mathcal{J}$, the lemma follows. For every $f^* \in S^*$, define $\text{nrst}(f^*) := \arg\min_{f \in S} d(f, f^*)$ be the closest facility in $S$ to $f^*$. For every client $v \in C$, define $\phi'(v) := \text{nrst}(\phi^*(v))$. See Figure 6 in the supplementary material for an illustrative example. The following two claims prove the lemma. □

**Claim 4.** $\phi'$ satisfies eq. (RD) and eq. (MP)

*Proof.* See Appendix B. □

**Claim 5.** $\mathcal{L}_p(S; \phi') \leq (\rho + 2) \text{OPT}_{\text{fair}}(\mathcal{I})$.

*Proof.* Fix a client $v \in C$. For the sake of brevity, let: $f = \phi(v)$, $f' = \phi'(v)$, and $f^* = \phi^*(v)$. We have

$$d(v, f') = d(v, \text{nrst}(f^*)) \leq d(v, f^*) + d(f^*, \text{nrst}(f^*)) \leq d(v, f^*) + d(f^*, f) \leq 2d(v, f^*) + d(v, f)$$

The first and third follows from triangle inequality while the second follows from the definition of nrst. Therefore, if we define the assignment cost vectors corresponding to $\phi$, $\phi'$, and $\phi^*$ as $\vec{d} = \{d(v, \phi) : v \in C\}$, $\vec{d'} = \{d(v, \phi') : v \in C\}$, and $\vec{d^*} = \{d(v, \phi^*) : v \in C\}$ respectively, the above equation implies $\vec{d'} \leq 2\vec{d} + \vec{d^*}$. Now note that the $\mathcal{L}_p$ is a monotone norm on these vectors, and therefore,

$$\mathcal{L}_p(S; \phi') = \mathcal{L}_p(\vec{d'}) \leq 2\mathcal{L}_p(\vec{d}) + \mathcal{L}_p(\vec{d^*}) = 2\mathcal{L}_p(S^*; \phi^*) + \mathcal{L}_p(S; \phi)$$

The proof is complete by noting $\mathcal{L}_p(S^*; \phi^*) = \text{OPT}_{\text{fair}}(\mathcal{I})$ and $\mathcal{L}_p(S; \phi) \leq \rho \cdot \text{OPT}_{\text{fair}}(\mathcal{I})$. □

### 3.2 Algorithm for the FAIR $p$-ASSIGNMENT problem

To complete the proof of Theorem 1, we need to give an algorithm for the FAIR $p$-ASSIGNMENT problem. We present this in Algorithm 1. The following theorem then establishes our main result.

**Theorem 6.** *There exists a $(4\Delta + 3)$-violating algorithm for the* FAIR $p$-ASSIGNMENT *problem.*

*Proof.* Fix an instance $\mathcal{J}$ of the problem. We start by writing a natural LP-relaxation[1].

$$\text{LP} := \min \sum_{v \in C, f \in S} d(v, f)^p x_{v,f} \qquad x_{v,f} \in [0, 1], \ \forall v \in C, f \in S \qquad \text{(LP)}$$

$$\beta_i \sum_{v \in C} x_{v,f} \ \leq \ \sum_{v \in C_i} x_{v,f} \ \leq \ \alpha_i \sum_{v \in C} x_{v,f} \qquad \forall f \in S, \forall i \in [\ell] \qquad \text{(1a)}$$

$$\sum_{f \in S} x_{v,f} \ = \ 1 \qquad \forall v \in C \qquad \text{(1b)}$$

**Claim 7.** $\text{LP} \leq \text{OPT}_{\text{asgn}}(\mathcal{J})^p$. □

Let $x^\star$ be an optimum solution to the above LP. Note that $x^\star$ could have many coordinates *fractional*. In Algorithm 1, we *iteratively round* $x^\star$ to an *integral* solution with the same or better value, but which violates the fairness constraints by at most $4\Delta + 3$. Our algorithm effectively simulates an algorithm for *minimum degree-bounded matroid basis* problem (MBDMB henceforth) due to Király et al. [39]. In this problem one is given a matroid $M = (X, \mathcal{I})$, costs on elements in $X$, a hypergraph $H = (X, \mathcal{E})$, and functions $f : \mathcal{E} \to \mathbb{R}$ and $g : \mathcal{E} \to \mathbb{R}$ such that $f(e) \le g(e)$ for all $e \in \mathcal{E}$. The objective is to find the minimum cost basis $B \subseteq X$ such that for all $e \in \mathcal{E}$, $f(e) \le |B \cap e| \le g(e)$. Now we state the main result in Király et al [39].

**Theorem 8** (Paraphrasing of Theorem 1 in [39]). *There exists a polynomial time algorithm that outputs a basis $B$ of cost at most* OPT*, such that $f(e) - 2\Delta_H + 1 \le |B \cap e| \le g(e) + 2\Delta_H - 1$ for each edge $e \in \mathcal{E}$ of the hypergraph, where $\Delta_H = max_{v \in X}|\{e \in E_H : v \in e\}|$ is the maximum degree of a vertex in the hypergraph $H$, and* OPT *is the cost of the natural LP relaxation.*

---

**Algorithm 1** Algorithm for the FAIR $p$-ASSIGNMENT problem

---

1: **procedure** FAIRASSIGNMENT$((\mathcal{X}, d), S, C = \cup_{i=1}^\ell C_i, \vec{\alpha}, \vec{\beta} \in [0, 1]^\ell)$
2:     $\hat{\phi}(v) = \emptyset$ for all $v \in C$
3:     solve the LP given in eq. (1), let $x^\star$ be an optimal solution
4:     for each $x^\star_{v,f} = 1$, set $\hat{\phi}(v) = f$ and remove $v$ from $C$ (and relevant $C_i$s).
5:     let $T_f := \sum_{v \in C} x^\star_{v,f}$ for all $f \in S$
6:     let $T_{f,i} := \sum_{v \in C_i} x^\star_{v,f}$ for all $i \in [\ell]$ and $f \in S$
7:     construct LP2 as given in eq. (2), only with variables $x_{v,f}$ such that $x^\star_{v,f} > 0$
8:     **while** there exists a $v \in C$ such that $\hat{\phi}(v) = \emptyset$ **do**
9:         solve LP2, let $x^\star$ be an optimal solution
10:         for each $x^\star_{v,f} = 0$, delete the variable $x^\star_{v,f}$ from LP2
11:         for each $x^\star_{v,f} = 1$, set $\hat{\phi}(v) = f$ and remove $v$ from $C$ (and relevant $C_i$s). Reduce $T_f$ and relevant $T_{f,i}$'s by 1.
12:         for every $i \in [\ell]$ and $f \in S$, if $|x^\star_{v,f} : 0 < x^\star_{v,f} < 1, \ v \in C_i| \le 2(\Delta + 1)$ remove the respective constraint in eq. (2c)
13:         for every $f \in S$, if $|x^\star_{v,f} : 0 < x^\star_{v,f} < 1, \ v \in C| \le 2(\Delta + 1)$ remove the respective constraint in eq. (2b)

---

$$\text{LP2} := \min \sum_{v \in C, f \in S} d(v, f)^p x_{v,f} \qquad\qquad x_{v,f} \in [0, 1], \ \ \forall v \in C, f \in S \qquad (2a)$$

$$\lfloor T_f \rfloor \ \le \ \sum_{v \in C} x_{v,f} \ \le \ \lceil T_f \rceil \qquad\qquad \forall f \in S, \forall i \in [\ell] \qquad (2b)$$

$$\lfloor T_{f,i} \rfloor \ \le \ \sum_{v \in C_i} x_{v,f} \ \le \ \lceil T_{f,i} \rceil \qquad \forall f \in S, \forall i \in [\ell] \qquad (2c)$$

$$\sum_{f \in S} x_{v,f} \ = \ 1 \qquad\qquad \forall v \in C \qquad (2d)$$

However, rather than posing our problem as an MBDMB instance, we write a natural LP-relaxation more suitable to the task — this is given in eq. (2). Proof of Theorem 6 is completed by drawing parallel to the Király et al [39] analysis for the MBDMB problem; the details are in Appendix B. $\square$

**Remark 1.** For the case of $p = \infty$, the objective function of eq. (LP) doesn't make sense. Instead, one proceeds as follows. We begin with a guess $G$ of $\text{OPT}_{\mathsf{asgn}}(\mathcal{J})$; we set $x_{v,f} = 0$ for all pairs with $d(v, f) > G$. We then check if eqs. (1a) and (1b) have a feasible solution. If they do not, then our guess $G$ is infeasible (too small). If they do, then the proof given above returns an assignment which violates eqs. (RD) and (MP) by additive $4\Delta + 3$, and satisfies $d(v, \phi(v)) \le G$ for all $v \in C$.

**Remark 2.** When $\Delta = 1$, that is, the $C_i$'s are disjoint, we can get an improved $+3$ additive violation (instead of $+7$). Instead of using Theorem 8, we use the generalized assignment problem (GAP) rounding technique by Shmoys and Tardos [52] to achieve this.

**Remark 3.** Is having a bicriteria approximation necessary? We do not know. The nub is the FAIR $p$-ASSIGNMENT problem. It is not hard to show that deciding whether a $\lambda$-violating solution exists with $\lambda = 0$ under the given definition is NP-hard. [2] However, an algorithm with $\lambda = 0$ and cost within a constant factor of $\mathrm{OPT}_{\mathsf{asgn}}(\mathcal{J})$ is not ruled out. This is an interesting open question.

## 4 Experiments

In this section, we perform empirical evaluation of our algorithm. We implement our algorithm in Python 3.6 and run all our experiments on a Macbook Air with a 1.8 GHz Intel Core i5 Processor and 8 GB 1600 MHz DDR3 memory. We use CPLEX[34] for solving LP's. Based on our experiments, we report four key findings: **(1)** Vanilla clustering algorithms are quite *unfair* even when measured against relaxed settings of $\alpha$ and $\beta$. In contrast, our algorithm's additive violation is almost always less than 3, even with $\Delta = 2$, across a wide range of parameters (see fig. 1 and fig. 2). **(2)** The cost of our fair clustering is close to the (unfair) vanilla cost for $k \leq 10$ as in fig. 3. In fact, we see (in fig. 7 in Appendix C.3) that our algorithm's cost is *very close* to the *absolute best* fair clustering that allows additive violations! Furthermore, our results for $k$-median significantly improve over the costs reported in Chierichetti et al. [19] and Backurs et al. [9] (see Table 1 ). **(3)** For the case of overlapping protected groups ($\Delta > 1$), enforcing fairness with respect to one sensitive attribute (say gender) can lead to unfairness with respect to another (say race). This empirical evidence stresses the importance of considering $\Delta > 1$ (see fig. 2 in Appendix C.4). **(4)** Finally, we study how the cost of our fair clustering algorithm changes with the strictness of the fairness conditions. This enables the user to figure out the trade-offs between fairness and utility and make an informed decision about which threshold to choose (see Appendix C.5).

**Datasets.** We use five datasets from the UCI repository [25]: [3] **(1)** `bank` [54] with 4,521 points, corresponding to phone calls from a marketing campaign by a Portuguese banking institution. **(2)** `census` [43] with 32,561 points, representing information about individuals extracted from the 1994 US census. **(3)** `diabetes` [53] with 101,766 points, extracted from diabetes patient records. **(4)** `creditcard` [33] with 30,000 points, related to information on credit card holders from a certain credit card in Taiwan. **(5)** `census1990` [47] with 2,458,285 points, taken from the 1990 US census, which we use for run time analysis. For each of the datasets, we select a set of numerical attributes to represent the records in the Euclidean space. We also choose two sensitive attributes for each dataset (e.g. sex and race for `census` ) and create protected groups based on their values. Appendix C.1 contains a more detailed description of the datasets and our features.

**Measurements.** For any clustering, we mainly focus on two metrics. One is the *cost of fairness*, that is, the ratio of the objective values of the fair clustering over the vanilla clustering. The other is *balance*, the measure of unfairness. To define balance, we generalize the notion found by Chierichetti et al. [19], We define two intermediate values $r_i$, the representation of group $i$ in the dataset and $r_i(f)$, the representation of group $i$ in cluster $f$ as $r_i := |C_i|/|C|$ and $r_i(f) := |C_i(f)|/|C(f)|$. Using these two values, balance is defined as $\mathrm{balance}(f) := \min\{r_i/r_i(f), r_i(f)/r_i\} \; \forall i \in [\ell]$. Although in theory the values of $\alpha, \beta$ for a given group $i$ can be set arbitrarily, in practice they are best set with respect to $r_i$, the ratio of the group in the dataset. Furthermore, to reduce the degrees of freedom, we parameterize $\beta$ and $\alpha$ by a single variable $\delta$ such that $\beta_i = r_i(1 - \delta)$ and $\alpha_i = r_i/(1 - \delta)$. Thus, we can interpret $\delta$ as how loose our fairness condition is. This is because $\delta = 0$ corresponds to each group in each cluster having exactly the same ratio as that group in the dataset, and $\delta = 1$ corresponds to no fairness constraints at all. For all of the experiments, we set $\delta = 0.2$ (corresponding to the common interpretation of the $80\%$-rule of DI doctrine), and use $\Delta = 2$, unless otherwise specified.

**Algorithms.** For vanilla $k$-center, we use a 2-approximation algorithm due to Gonzalez [30]. For vanilla $k$-median, we use the single-swap 5-approximation algorithm by Arya et al. [8], augment it with the $D$-sampling procedure by [7] for initial center section, and take the best out of 5 trials. For $k$-means, we use the $k$-means++ implementation of [48].

**Fairness comparison with vanilla clustering.** In fig. 1 we motivate our discussion of fairness by demonstrating the unfairness of vanilla clustering and fairness of our algorithm. On the $x$-axis, we compare three solutions: (1) our algorithm (labelled "ALG"), (2) fractional solution to the FAIR $p$-ASSIGNMENT LP in Equation (1) (labelled "Partial"), and (3) vanilla $k$-means (labelled "VC").

Below these labels, we record the *cost of fairness*. We set $\delta = 0.2$ and $k = 4$. Along the $y$ axis, we plot the balance metric defined above for the three largest clusters for each of these clustering. The dotted line at $0.8$ is the goal balance for $\delta = 0.2$. The lowest balance for any cluster for our algorithm is $0.75$ (for census ), whereas vanilla can be as bad as $0$ (for bank ); "partial" is, of course, always fair (at least $0.8$). We observe that the maximum additive violation of our algorithm is only $3$ (much

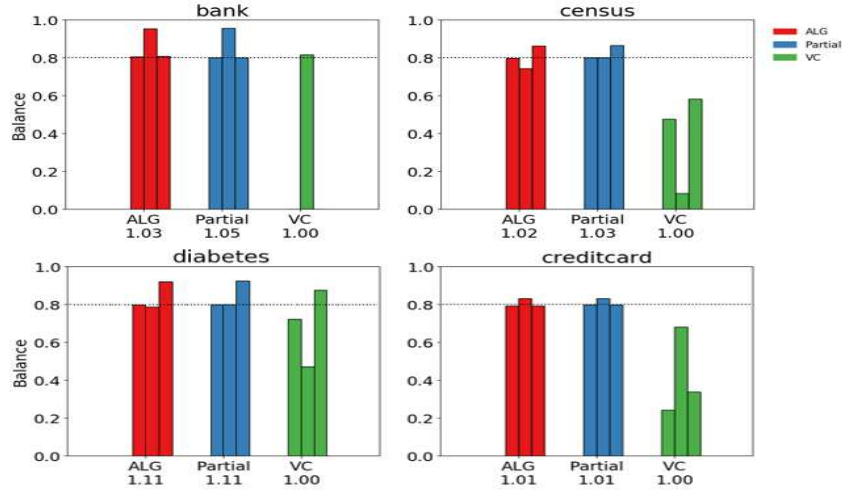

Figure 1: Comparison of our algorithm (ALG) versus vanilla clustering (VC) in terms of balance for the $k$-means objective.

better than our theoretical bound of $4\Delta + 3$)), for a large range of values of $\delta$ and $k$, whereas vanilla $k$-means can be unfair by quite a large margin. (see fig. 2 below and Table 3 in Appendix C.2).

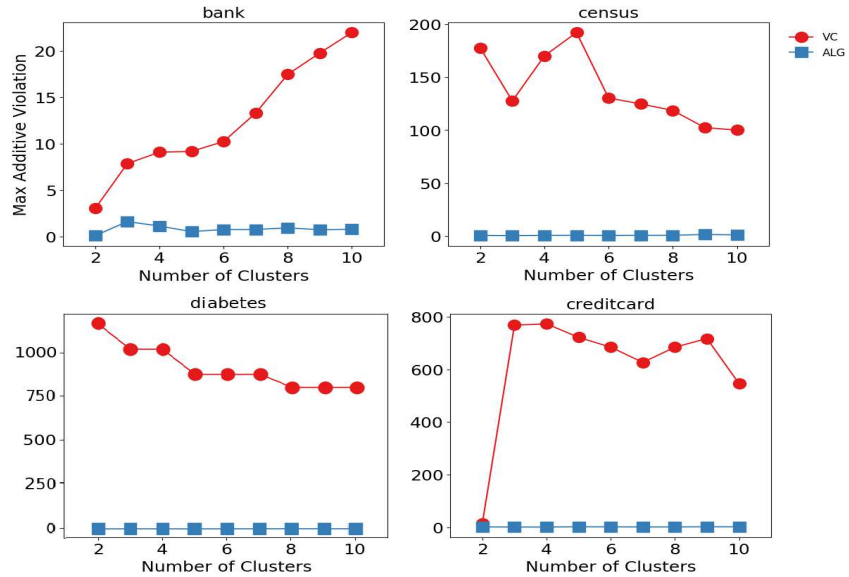

Figure 2: Comparison of the maximum additive violation (for $\delta = 0.2$ and $\Delta = 2$) over all clusters and all groups between our algorithm (ALG) and vanilla (VC), using the $k$-means objective.

**Cost analysis.** We evaluate the cost of our algorithm for $k$-means objective with respect to the vanilla clustering cost. Figure 3 shows that the cost of our algorithm for $k \leq 10$ is at most 15% more than the vanilla cost on bank , census , and creditcard . Interestingly, for creditcard , even though the vanilla solution is extremely unfair as demonstrated earlier, cost of fairness is at most 6% which indicates that the vanilla centers are in the "right place".

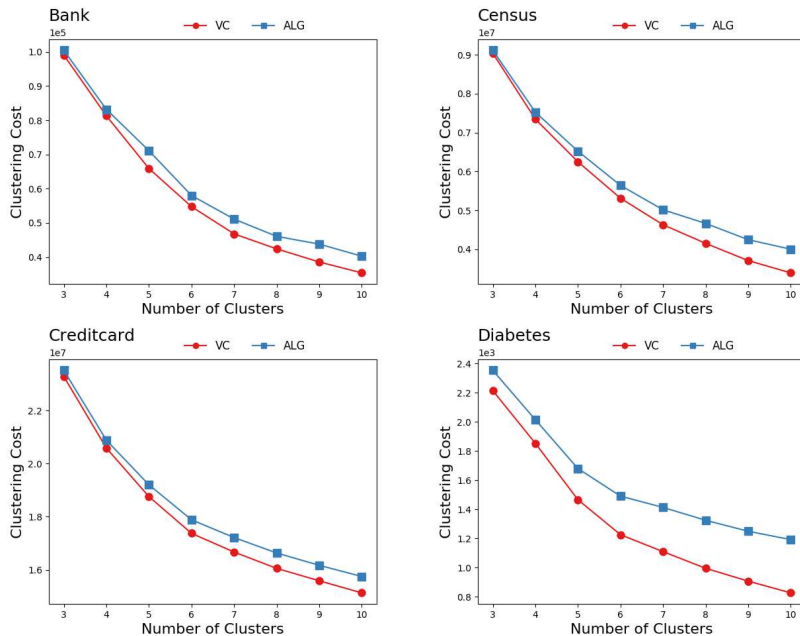

Figure 3: Our algorithm's cost (ALG) versus the vanilla clustering cost (VC) for $k$-means objective.

Our results in Table 1 confirm that we outperform [19] and [9] in terms of cost. To match [19] and [9], we sub-sample `bank`, `census`, and `diabetes` to 1000, 600, and 1000 respectively, declared only one sensitive attribute for each (i.e. marital for `bank`, sex for `census`, and gender for `diabetes`), and tune the fairness parameters to enforce a balance of $0.5$. The data in table 1 for [9] is the output of their code, and the numbers for [19] are drawn from their plots.

Table 1: Comparison of our clustering cost with [9] and [19] for $k$-median with varying $k$.

| | k | 3 | 4 | 5 | 6 | 7 | 8 | 9 | 10 |
|---|---|---|---|---|---|---|---|---|---|
| census<br>cost $\times 10^{-6}$ | **Ours** | 19.55 | 16.63 | 14.35 | 11.75 | 9.86 | 8.87 | 7.75 | 7.32 |
| | [9] | 28.29 | 28.57 | 26.31 | 22.21 | 24.81 | 26.94 | 20.80 | 23.60 |
| | [19] | 40 | 39 | 38.5 | 38 | 37.8 | 37.75 | 37.6 | 37.5 |
| bank<br>cost $\times 10^{-5}$ | **Ours** | 6.81 | 5.64 | 4.95 | 4.49 | 4.05 | 3.79 | 3.53 | 3.44 |
| | [9] | 8.05 | 7.78 | 7.65 | 6.63 | 6.33 | 6.68 | 5.42 | 6.70 |
| | [19] | 5.9 | 5.8 | 5.77 | 5.75 | 5.7 | 5.65 | 5.62 | 5.6 |
| diabetes<br>cost | **Ours** | 6675 | 5491 | 3890 | 3371 | 3194 | 2939 | 2700 | 2380 |
| | [9] | 7756 | 6412 | 5526 | 4746 | 4850 | 4765 | 4203 | 4337 |
| | [19] | 11500 | 10300 | 10250 | 10200 | 10175 | 10150 | 10125 | 10100 |

**Run time analysis.** In this paper, we focus on providing a framework and do not emphasize on run time optimization. Nevertheless, we note that our algorithm for the k-means objective finds a fair solution for the `census1990` dataset with 500K points and 13 features in less than 30 minutes (see Table 2). Even though our approach is based on iterative rounding method, in practice CPLEX solution to LP (eq. (1)) is more than 99% integral for each of our experiments. Hence, we never have to solve more than two or three LP. Also the number of variables in subsequent LPs are significantly small. In contrast, if we attempt to frame LP (eq. (1)) as an integer program instead, the CPLEX solver fails to find a solution in under an hour even with 40K points.

Table 2: Runtime of our algorithm on subsampled data from `census1990` for $k$-means ($k = 3$).

| Number of sampled points | 10K | 50K | 100K | 200K | 300K | 400K | 500K |
|---|---|---|---|---|---|---|---|
| Time (sec) | 4.04 | 33.35 | 91.15 | 248.11 | 714.73 | 1202.89 | 1776.51 |

## Footnotes

[1]This makes sense only for finite $p$. See Remark 1

[2]A simple reduction from the 3D-matching problem.

[3]https://archive.ics.uci.edu/ml/datasets/

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
