[Supplementary Material · Fair_Clustering_Appendix.pdf]

# A  Our algorithmic template

In this section we present our algorithmic template for the FAIR $(k, p)$-CLUSTERING problem in Algorithm 2. This template uses the FAIRASSIGNMENT procedure ( Algorithm 1) as a subroutine.

---

**Algorithm 2** Algorithm for the FAIR $(k, p)$-CLUSTERING problem

---

1: **procedure** FAIRCLUSTERING$((\mathcal{X} = F \cup C, d), C = \cup_{i=1}^{\ell} C_i, \vec{\alpha}, \vec{\beta} \in [0, 1]^{\ell})$
2:     solve the VANILLA $(k, p)$-CLUSTERING problem on $(\mathcal{X}, d)$
3:     let $(S, \phi)$ be the solution
4:     $\hat{\phi}$ = FAIRASSIGNMENT $((\mathcal{X}, d), S, C = \cup_{i=1}^{\ell} C_i, \vec{\alpha}, \vec{\beta})$ ( Algorithm 1)
5:     return $(S, \hat{\phi})$

---

# B  Missing proofs from Section 3

Figure 4                                    Figure 5

Figure 6: In this figure, we demonstrate various components of our proof. On the left, in fig. 4, we give an example of the assignment functions $\phi$, $\phi'$, and $\phi^\star$. The client $v$ is assigned to the facility $f$ in the vanilla solution $S$. In the fair optimal solution, it is assigned to the facility $f^\star$ and $f'$ is the nearest facility in $S$ to $f^\star$. On the right, in Figure 5, we show the crux of the proof of Claim 5.

*Proof of Claim 4.* For any facility $f^* \in S^*$, let $C(f^*) := \{v : \phi^*(v) = f^*\}$. The $C(f^*)$'s partition $C$. For any $i \in [\ell]$, let $C_i(f^*) := C(f^*) \cap C_i$. Since $(S^*; \phi^*)$ is a feasible solution satisfying the fairness constraints, we get that for every $f^* \in S^*$ and for every $i \in [\ell]$, $\beta_i \leq \frac{|C_i(f^*)|}{|C(f^*)|} \leq \alpha_i$.

For any facility $f \in S$, let $N(f) := \{f^* \in S^* : \mathsf{nrst}(f^*) = f\}$ be all the facilities in $S^*$ for which $f$ is the nearest facility. Note that the clients $\{v \in C : \phi'(v) = f\}$ are precisely $\dot\cup_{f^* \in N(f)} C(f^*)$. Similarly, for any $i \in [\ell]$, we have $\{v \in C_i : \phi'(v) = f\}$ is precisely $\dot\cup_{f^* \in N(f)} C_i(f^*)$. Therefore, $\frac{|\{v \in C_i : \phi'(v) = f\}|}{|\{v \in C : \phi'(v) = f\}|} = \frac{\sum_{f^* \in N(f)} |C_i(f^*)|}{\sum_{f^* \in N(f)} |C(f^*)|} \in [\beta_i, \alpha_i]$ since the second summation is between $\min_{f^* \in N(f)} |C_i(f^*)|/|C(f^*)|$ and $\max_{f^* \in N(f)} |C_i(f^*)|/|C(f^*)|$, and both these are in $[\beta_i, \alpha_i]$. □

*Proof of Theorem 6.* Recall $x^\star$ is an optimum solution to the LP given in eq. (1). To prove the theorem, we first construct an instance of the MBDMB problem using $x^\star$. Then we appeal to Theorem 8 to argue about the quality of our algorithm.

Let $E$ be the set of $(v, f)$ pairs with $x^\star_{v,f} > 0$. For a point $v \in C$, let $E_v$ denote the set of edges in $E$ incident on $v$. Define $\mathcal{F} := \{F \subseteq E : |F \cap E_v| \leq 1 \ \forall v \in C\}$ to be collection of edges which "hit" every client at most once. The pair $M = (E, \mathcal{F})$ is a well known combinatorial object called a (partition) *matroid*. For each element $(v, f)$ of this matroid $M$, we denotes its cost to be $c(v, f) := d(v, f)^p$.

Next we define a *hypergraph* $H = (E, \mathcal{E})$. For each $f \in S$ and $i \in [\ell]$, let $E_{f,i} \subseteq E$ consisting of pairs $(v, f) \in E$ for $v \in C_i$. Let $E_f := \cup_{i=1}^{\ell} E_{f,i}$. Each of these $E_{f,i}$'s and $E_f$'s are added to the collection of hyperedges $\mathcal{E}$. Next, let $T_f := \sum_{v \in C} x^\star_{v,f}$ be the total fractional assignment on $f$. Similarly, for all $i \in [\ell]$, define $T_{f,i} := \sum_{v \in C_i} x^\star_{v,f}$. Note that, both $T_f$ and $T_{f,i}$ can be fractional. For every $e \in E_{f,i}$, we define $f(e) := \lfloor T_{f,i} \rfloor$ and $g(e) = \lceil T_{f,i} \rceil$. For each $e \in E_f$, we denote $f(e) = \lfloor T_f \rfloor$ and $g(e) = \lceil T_f \rceil$. This completes the construction of the MBDMB instance.

Now we can apply Theorem 8 to obtain a basis $B$ of matroid $M$ with the properties mentioned. Note that for our hypergraph $\Delta_H \leq \Delta + 1$ where $\Delta$ is the maximum number of groups a client can be in. This is because every pair $(v, f)$ belongs to $E_f$ and $E_{f,i}$'s for all $C_i$'s containing $v$. Also note that any basis corresponds to an

assignment $\phi : C \to S$ of all clients. Furthermore, the cost of the basis is precisely $\mathcal{L}_p(S; \phi)^p$. Since this cost is $\leq \text{LP} \leq \text{OPT}_{\text{fair}}(\mathcal{J})^p$, we get that $\mathcal{L}_p(S; \phi) \leq \text{OPT}_{\text{fair}}(\mathcal{J})$. We now need to argue about the violation.

Fix a server $f$ and a client group $C_i$. Let $\overline{T}_f$ and $\overline{T}_{f,i}$ denote the number of clients assigned to $f$ and the number of clients from $C_i$ that are assigned to $f$ respectively (by the integral assignment). Then, by Theorem 8, $\lfloor T_f \rfloor - 2\Delta - 1 \leq \overline{T}_f \leq \lceil T_f \rceil + 2\Delta + 1$ and $\lfloor T_{f,i} \rfloor - 2\Delta - 1 \leq \overline{T}_{f,i} \leq \lceil T_{f,i} \rceil + 2\Delta + 1$ (using $\Delta_H \leq \Delta + 1$). Now consider eq. (RD). Since, $T_{f,i} \leq \alpha_i T_f$ (as the LP solution is feasible),

$$\overline{T}_{f,i} \leq \lceil \alpha_i T_f \rceil + 2\Delta + 1 \leq \alpha_i \lfloor T_f \rfloor + 2\Delta + 2 \leq \alpha_i (\overline{T}_f + 2\Delta + 1) + 2\Delta + 2 \leq \alpha_i \overline{T}_f + (4\Delta + 3),$$

where the second last inequality follows as $\alpha_i \leq 1$. We can similarly argue about eq. (MP). This completes the proof of Theorem 6. □

# C    More experimental results

In this section, we present additional experimental evaluations of our algorithm.

## C.1    Details of the Datasets

We present the details about the features and sensitive attributes of the datasets used in our algorithms in Table 3.

Table 3: For each dataset, the coordinates are the numeric attributes used to determined the position of each record in the Euclidean space. The sensitive attributes determines protected groups.

| Dataset | Coordinates | Sensitive attributes | Protected groups |
|---|---|---|---|
| bank | age, balance, duration | marital | married, single, divorced |
| | | default | yes, no |
| census | age, education-num, final-weight, capital-gain, hours-per-week | sex | female, male |
| | | race | Amer-ind, asian-pac-isl, black, other, white |
| diabetes | gender, age, race, time-in-hospital | gender | female, male |
| | | race | 6 groups |
| creditcard | age, bill-amt 1 — 6, limit-bal, pay-amt 1 — 6 | marriage | married, single, other, null |
| | | education | 7 groups |
| census1990 | dAncstry1, dAncstry2, iAvail, iCitizen, iClass, dDepart, iFertil, iDisabl1, iDisabl2, iEnglish, iFeb55, dHispanic, dHour89 | dAge | 8 groups |
| | | iSex | female, male |

## C.2    Maximum additive violations of our algorithm

For a wide range of values of $\delta$ and $k$, we never violate the fairness constraints by more that an additive amount of $3.02$. In comparison, the vanilla $k$-means violates fairness by quite a large margin. Note that that setting of $\delta = 0.2$ corresponds to the common interpretation of $80\%$ rule of the DI doctrine. We give a detailed report in Table 4.

## C.3    Additional cost analysis

We first, evaluate the cost of our algorithm for $k$-means objective with respect to the vanilla clustering cost and the *almost fair LP* cost. The almost fair LP (eq. (3)) is an LP relaxation of FAIR $(k, p)$-CLUSTERING, with variables for choosing the centers, except that we allow for a $\lambda$ additive violation in fairness. The cost of this LP is a lower-bound on the cost of *any* fair clustering that violates fairness by at most an additive factor of $\lambda$.

Table 4: The maximum additive violation across a range of $\delta$ of our algorithm compared to vanilla $k$-means. For each $\delta$, we take maximum over $k$, for $k \in [2, 10]$ on all datasets.

| $\delta$ | 0.01 | 0.05 | 0.1 | 0.2 | 0.3 | 0.4 | 0.5 | Vanilla ($\delta = 0.2$) |
|---|---|---|---|---|---|---|---|---|
| bank | 1.45 | 1.17 | 1.39 | **1.54** | 1.19 | 1.15 | 1.03 | **21.99** |
| census | 1.44 | 1.53 | **1.89** | 1.08 | 1.18 | 0.97 | 1.03 | **773.19** |
| creditcard | **3.02** | 2.32 | 2.11 | 2.29 | 2.03 | 1.63 | 1.03 | **192.01** |

$$\text{LP3} := \min \sum_{v \in C, f \in S} d(v,f)^p x_{v,f} \qquad x_{v,f} \in [0,1], \ \forall v \in C, f \in S \tag{3a}$$

$$\sum_{f \in S} x_{v,f} = 1 \qquad \forall v \in C \tag{3b}$$

$$x_{v,f} \leq y_f \qquad \forall v \in C, f \in S \tag{3c}$$

$$\sum_{f \in S} y_f \leq k \tag{3d}$$

$$\sum_{v \in C_i} x_{v,f} \leq \alpha_i \sum_{v \in C} x_{v,f} + \lambda \qquad \forall f \in S, \forall i \in [\ell] \tag{3e}$$

$$\sum_{v \in C_i} x_{v,f} \geq \beta_i \sum_{v \in C} x_{v,f} - \lambda \qquad \forall f \in S, \forall i \in [\ell] \tag{3f}$$

In fig. 7 we compare the cost of our algorithm with a lower-bound on the absolute best cost of any clustering that has the same amount of violation as ours. To be more precise, for any dataset we set $\lambda$ according to the maximum violation of our algorithm reported in table 4 for $\delta = 0.2$ (e.g. $\lambda$ is 1.54 for bank , 1.08 for census , and 2.29 for creditcard ). Then, we solve the almost fair LP for that $\lambda$ and compare its cost with our algorithm's cost over that dataset.

Since solving the almost fair LP on the whole data is infeasible (in terms of running time), we sub-sample bank , census , and creditcard to 1000, 600, and 600 points respectively, and report the average costs over 10 trials. Also, we only consider one sensitive attribute, namely marital for bank , sex for census , and education for creditcard to further simplify the LP and decrease the running time. fig. 3 shows that the cost of our algorithm is very close to the almost fair LP cost (at most 15% more). Note that, since the cost of almost fair LP is a lower bound on the cost of FAIR $(k, p)$-CLUSTERING problem, we conclude that our cost is at most 15% more than the optimum in practice, which is much better than the proved $(\rho + 2)$ factor in theorem 1.

Figure 7: Average costs of vanilla clustering (VC), our algorithm (ALG), and almost fair LP (AFLP), for $k$-means objective, as a function of $k$.

## C.4 The case of $\Delta > 1$

In this section, we demonstrate the importance of considering $\Delta > 1$ by showing that enforcing fairness with respect to one attribute (say gender) may lead to significant unfairness with respect to another attribute (say race). In Figure 8, we have two plots for each dataset. In each plot, we compare three clustering: (1) Our algorithm with $\Delta = 2$ (labelled "both"); (2) and (3) Our algorithm with $\Delta = 1$ with protected groups defined by the attribute on $x$-axis label. We set $\delta = 0.2$ and $k = 4$. The clustering objective is $k$-means. Along $y$-axis, we measure the *balance* metric for the three largest clusters for each of these clustering. In each plot we only measure the *balance* for the attribute written in bold in the top right corner.

In datasets, such as `bank`, we see that fairness with respect to only the marital attribute leads to a large amount of unfairness in the default attribute. The fairest solution along both attributes is when they are both considered by our algorithm ($\Delta = 2$). Interestingly, there are datasets where fairness by one attribute is all that is needed. On the `census` dataset, fairness by race leads to a fair solution on sex, but fairness by sex leads to large amount of unfairness in race.

Finally, our results strongly suggest that finding a fair solution for two attributes is often only slightly more expensive (in terms of the clustering objective) than finding a fair solution for only one attribute.

Figure 8: Importance of considering $\Delta > 1$. Below these x labels is the *cost of fairness* ratio. We report the *balance* for the three largest clusters and include the dotted line at $0.8$ because we use $\delta = 0.2$.

## C.5 Tuning the fairness parameters

In Figure 9, we demonstrate the ability to tune the strictness of the fairness criteria by manipulating the parameter $\delta$. As $\delta$ approaches $1$, the ratio between the fair objective and original vanilla objective decreases to $1$. This suggests that the fair solution has recapitulated the vanilla clustering because our bounds are lax enough to do so.

Figure 9: We show the effects of varying $\delta$ (x-axis) on our algorithm's fair objective cost over the vanilla cost (y-axis).