[Reviews · NeurIPS 2019]

Reviewer 1



The paper is interesting, well-written and addresses an important topic. Its main contributions are described in the section above. This paper is original in addressing a generalization of previously considered fair clustering cases (one of them published in NIPS 2017). As mentioned above, the authors note that a similar generalization was considered in a parallel work (currently available on Arxiv). While the main LP-based algorithm is relatively simple and uses quite standard methods (effectively MBDMB), it is not trivial, and these methods are used for this problem in a nice way. The experimental results are also interesting, but using just 4 UCI datasets for studying this general problem seems too little, and might not reflect the overall behavior. It is unclear, for example, why the authors did not use the Diabetes dataset used in the NIPS 2017 paper of Chierichetti et al that they cite. The main significance of the paper is in the theoretical result, which significantly improves upon previous work, while the experiments mainly explore sample cases and demonstrate that they may behave better than guaranteed. I am currently grading this paper as a good submission. For accepting it following the authors response, I expect to at least see experimental results for the missing dataset out of the datasets considered in the NIPS2017 paper (UCI Diabetes).

Reviewer 2



The paper studies fair variants of a class of clustering problems that include k-center, k-median, and k-means. The fairness constraint studied here is a generalization of the one introduced by Chierchietti et al. The main result is a black-box reduction of the fair clustering problem to the vanilla clustering problem. This is an interesting result, and immediately gives constant-factor approximation algorithms (with constant violation of the fairness constraint) for the corresponding fair clustering problems. The experimental results are not impressive. Mainly, the fair algorithm given in the paper is compared with the vanilla solution. No comparison with a baseline is provided. Also, the paper does not discuss a recent related paper by Ahmadian et al. (in KDD 2019). The two papers have some overlap in the results and the techniques. Even though the overlap doesn't cover the main result of the present paper (the black-box reduction), it still needs to be discussed.

Reviewer 3



The paper gives a new algorithm for a generalized version of the fair clustering problem. The algorithm works by using a black-box clustering algorithm and then reassigning points in order to meet the fairness constraints. The point reassignment algorithm is an LP relaxation solved via iterative rounding. The reduction and black-box setup of the algorithm is elegant and nicely simple. Because of the black-box clustering algorithm setup, this solution applies to k-means, k-median, and k-center algorithms.

[Author Response · NeurIPS 2019]

We thank all the reviewers for their very useful comments on our paper. To recall, our paper gives a simple theoretical framework for converting vanilla clustering algorithms into fair algorithms with a slight loss in performance, for any norm. Empirically, our algorithm outperforms known results and theoretical guarantees.

A common criticism across the reviews is on the experimental analysis. While we broadly agree on some of the points raised, we take this opportunity to address some of the comments. Reviewer 1 rightly points out that 4 UCI datasets is too low. Indeed, it is; however, it is not easy to find datasets where certain features are sensitive. We mostly used datasets that previous authors have been using. As the reviewer mentions, we didn't report the diabetes dataset. We are sorry for this oversight, and present the results below (table and the two figures on the right). The second issue was on benchmarks (Reviewer 2 brought this up); we would like to point out that we do compare our algorithm's performance with (a) previous algorithm's (Table 1), and also (b) with very stringent benchmarks ( Fig 4 in the supplemental material): finding the true optimum for the fair problem is hard; we actually compare ourselves with a *lower bound* (the LP solution) on the optimal fair solution, and show we are comparable on various datasets. Reviewer 3 asks why $k = 4$ was chosen, and why we only show three bar-charts. For these data-sets, the elbow-method does not quite indicate which $k$ to use. At some level, the choice of $k = 4$ is therefore arbitrary; we used one which illustrated our point the best. The reason we show three clusters is two-fold: (a) aesthetics, 12 bar charts seemed clunky, and (b) the fourth cluster was too small. Reviewer 3 also asks why we compared with the Backurs et al. [ICML 2019] paper for only $k = 20$. At the time of submission, the code of Backurs et al. was not available. Their paper only reported $k = 20$. Since then, their code has been made available, and in the table below we show a comparison for varying $k$ with the Chierichetti et. al. results drawn from the plot in their NeurIPS 2017 paper.

We now clarify some other reviewer questions. Reviewer 1 asks a very pertinent question about tightness/hardness. We give two answers. (a) Our algorithm is tight, in that running on the example shown in the figure below for $k$-center objective, our approximation factor is indeed 5. (b) The Fair-$p$-assignment problem, without additive violation, is also NP-hard. This follows from a simple reduction from 3D matching. However, we *do not know* of any hardness for the instances of Fair-$p$-assignment which arise from our reduction. So we cannot claim a general hardness. If accepted, we will definitely add a para discussing this. Reviewer 2 points out the recent related KDD 2019 paper by Ahmadian et. al. This paper became public (May 29th, on arXiv) only after the NeurIPS deadline . Indeed, their paper also studies the *restricted dominance*, but work for only $k$-center objective. Ours, at price of the additive violation, works for all norms. We will definitely add a comparison in the final version of our paper. Reviewer 3 has trouble understanding why the additive violation was needed, and whether simply fiddling with alpha's and beta's would work. In practice, maybe. But, imagine the following scenario – we wish to run for beta = 0.2 and alpha = 0.8; but keeping the violation in mind, we run with beta = 0.25 and alpha = 0.75. The issue is that our guarantee will compare with the optimal solution for this new (0.25,0.75) setting, which could be *larger*. Reviewer 3 also points out a picture would have been useful for Claim 5. We all had a smile on our face, because we had added a picture, but the page-limit forced to take it out. Perhaps, we should have added it to the supplementary material. The same reason holds for the "Conclusion" section. If accepted, we will take a hard look at how to save space so as to incorporate these comments above.

| | k-median | 3 | 4 | 5 | 6 | 7 | 8 | 9 | 10 |
|---|---|---|---|---|---|---|---|---|---|
| `census` , cost $\times 10^6$ | Ours | 19.55 | 16.63 | 14.35 | 11.75 | 9.86 | 8.87 | 7.75 | 7.32 |
| | Backurs et.al. | 28.29 | 28.57 | 26.31 | 22.21 | 24.81 | 26.94 | 20.80 | 23.60 |
| | Chierichetti et.al. | 40 | 39 | 38.5 | 38 | 37.8 | 37.75 | 37.6 | 37.5 |
| `bank` , cost $\times 10^5$ | Ours | 6.81 | 5.64 | 4.95 | 4.49 | 4.05 | 3.79 | 3.53 | 3.44 |
| | Backurs et.al. | 8.05 | 7.78 | 7.65 | 6.63 | 6.33 | 6.68 | 5.42 | 6.70 |
| | Chierichetti et.al. | 5.9 | 5.8 | 5.77 | 5.75 | 5.7 | 5.65 | 5.62 | 5.6 |
| `diabetes` | Ours | 6675 | 5491 | 3890 | 3371 | 3194 | 2939 | 2700 | 2380 |
| | Backurs et.al. | 7756 | 6412 | 5526 | 4746 | 4850 | 4765 | 4203 | 4337 |
| | Chierichetti et.al. | 11500 | 10300 | 10250 | 10200 | 10175 | 10150 | 10125 | 10100 |

Squares: facilities. Circles: clients (red or blue). All distances 1. $k = 4$. Optimum fair solution with perfect balance opens $\{a, c, e, g\}$ with cost 1. Our algorithm may open $\{a, e, f, h\}$ leading to cost 5.



[Meta-Review · NeurIPS 2019]

This paper gives a general reduction for converting unfair clustering solution for partition based clusterings (k-{center, median, means}) to a fair version of these. It's a timely and well executed work.